# Microbiome and Its Dysbiosis in Inborn Errors of Immunity

**DOI:** 10.3390/pathogens12040518

**Published:** 2023-03-27

**Authors:** Madhubala Sharma, Manpreet Dhaliwal, Rahul Tyagi, Taru Goyal, Saniya Sharma, Amit Rawat

**Affiliations:** AllergyImmunology Unit, Department of Pediatrics, Advanced Pediatrics Center, Postgraduate Institute of Medical Education and Research, Chandigarh 160012, India

**Keywords:** dysbiosis, inborn errors of immunity, primary immunodeficiency diseases, PIDs, microbes

## Abstract

Inborn errors of immunity (IEI) can present with infections, autoimmunity, lymphoproliferation, granulomas, and malignancy. IEIs are due to genetic abnormalities that disrupt normal host-immune response or immune regulation. The microbiome appears essential for maintaining host immunity, especially in patients with a defective immune system. Altered gut microbiota in patients with IEI can lead to clinical symptoms. Microbial dysbiosis is the consequence of an increase in pro-inflammatory bacteria or a reduction in anti-inflammatory bacteria. However, functional and compositional differences in microbiota are also involved. Dysbiosis and a reduced alpha-diversity are well documented, particularly in conditions like common variable immunodeficiency. Deranged microbiota is also seen in Wiskott–Aldrich syndrome, severe combined immunodeficiency, chronic granulomatous disease, selective immunoglobulin-A deficiency, Hyper IgE syndrome (HIGES), X-linked lymphoproliferative disease-2, immunodysregulation, polyendocrinopathy, enteropathy, x-linked syndrome, and defects of IL10 signalling. Distinct gastrointestinal, respiratory, and cutaneous symptoms linked to dysbiosis are seen in several IEIs, emphasizing the importance of microbiome identification. In this study, we discuss the processes that maintain immunological homeostasis between commensals and the host and the disruptions thereof in patients with IEIs. As the connection between microbiota, host immunity, and infectious illnesses is better understood, microbiota manipulation as a treatment strategy or infection prevention method would be more readily employed. Therefore, optimal prebiotics, probiotics, postbiotics, and fecal microbial transplantation can be promising strategies to restore the microbiota and decrease disease pathology in patients with IEIs.

## 1. Introduction

Primary immunodeficiency diseases (PIDs), now known as inborn errors of immunity (IEI), comprise 485 different clinical phenotypes associated with single gene defects in 485 genes [1]. IEI present with varied clinical manifestations, such as recurrent infections, autoimmunity, lymphoproliferation, malignancy, and granulomas [2]. IEI are a group of diverse diseases that range in severity, comorbidities, and genetic etiologies [3]. Recently, the microbiome has been considered an essential regulator of these diseases, with accumulating evidence indicating connections between altered gut microbiota and clinical symptoms in patients with different forms of IEI [4,5].Gut microbiota has been studied in some IEI, especially antibody defects such as common variable immunodeficiency (CVID), X-linked agammaglobulinemia (XLA), and selective immunoglobulin-A deficiency (sIgAD) [6]. Immunodeficiencies, such as severe combined immunodeficiency (SCID), Wiskott–Aldrich syndrome (WAS), and chronic granulomatous disease (CGD) can also lead to a dysbiotic state in these patients [6]. Humoral immunodeficiencies due to defects in B cell development or function are the most common form of IEI worldwide; therefore, studies on the association of microbiota with defective humoral immunity are a major focus of research. The microbiome appears essential for maintaining host immunity, especially in patients with a defective immune system as described in Figure 1 [7].

Human microbiota comprises multitudes of microbes which are in constant cross-talk withthe host immune system and shape the immune repertoire [8]. These living microorganisms, which include bacteria, viruses, fungi, and protozoa, make up commensal microflora and play an essential role in regulating host immunity [9]. Immunological mediators may be intrinsic (cytokines, microRNA, and the microbiome) or extrinsic (therapy regimens such as intravenous immunoglobin (IVIG), biologics, antibiotics, and nutrition).

Furthermore, smoking, alcohol use, and food are examples of personal microenvironmental factors that can affect health, and there are also contextual macroenvironmental factors that might affect health, such as the constructed environment and socioeconomic surroundings [10]. Studies have reported geographical and ethnic variations in the gut microbiota. Significant enrichment of *Prevotella* was seen in India, but in Japan increased abundance of *Bifidobacterium* and *Clostridium* were reported. On the other hand, increased prevalence of *Ruminococcus*, *Roseburia*, and *Veillonellaceae* was reported in the Dutch ethnicity [11]. Thus, microbiota is dynamic in nature and is influenced by several factors not limited to, diet, geographical location, ethnicity, and lifestyle.

## 2. Aging and Microbiota

Human gut microbiota is dynamic and varies with age. The mean species diversity of a place at a local scale is known as alpha diversity. There is no global difference between adults and children in beta-diversity (ratio between regional and local species diversity), but alpha diversity is lower in children as compared to adults. Microbiota composition in children includes higher diversity of the phyla Actinobacteria, Bacilli, *Ruminococcaceae*, and Bacteroidetes and a lower diversity of phyla Methanobacteriales compared to adults [12].

Aging can change the composition and diversity of the gut microbiome. This is due to a combination of factors, including changes in diet, medications, and lifestyle as well as changes in the immune system and other physiological functions. These changes can lead to a reduction in the beneficial microbes that help maintain gut health and function while increasing the abundance of harmful microbes [13]. The dysbiosis that occurs with aging has been linked to several age-related health problems, such as inflammation, immune dysfunction, and metabolic disorders. Older adults with lower levels of beneficial gut bacteria are more likely to have age-related chronic diseases such as type 2 diabetes, heart disease, and cognitive decline [14].

Age-related changes in the immune system and gut microbiome result in a greater propensity to contract infectious diseases and a decreased ability to respond to vaccinations. Addressing age-related dysbiosis may improve health and longevity by reducing systemic low-grade inflammation and immunosenescence—two hallmarks of aging [15].

To better understand the development of gut microbiota, a large cohort of newborns (*n* = 903) from Germany, Finland, Sweden, and the United States were examined and the microbiota assessed. It was seen that gut microbiota develops in three separate phases and varies based on alpha-diversity and the interplay amongst the phyla constituting the microbiota. The gut microbiota evolves as follows: (i) a period of development (3–14 months of life) during which the alpha-diversity and identified phyla steadily shift, (ii) a phase of transition (15–30 months of life) during which alpha-diversity continues to alter and Bacteroidetes and Proteobacteria continue to grow, and (iii) a stable period (≥31 months) during which the alpha-diversity and phyla present are unaltered. The stable phase is diverse and has a marked predominance of Firmicutes in contrast to the developing phase which is dominated by *Bifidobacterium* spp. These results suggest that the first three years of life are crucial for the establishment, composition, and function of the gut microbiota [12,16].

Accumulating scientific evidence indicates that loss of microbial diversity, or dysbiosis, may be related to inflammatory and immunological dysregulation processes in IEI [17]. In patients with IEIs, the microbiome plays a vital role in regulating inflammation, immune exhaustion, and immunodeficiency [18]. The biological factors, along with other conditions, including genetics, immune dysfunction, antibiotics, and diet, can cause inflammation, autoimmunity, lymphoproliferation, and risk of malignancy, which may affect the quality of life in these patients [19]. In this review, we will focus on dysbiosis in IEI.

## 3. Influence of Diet on Microbiome

In addition to genetics, environment, and medication use, diet plays a significant role in determining the microbiome.Research has shown that certain dietary patterns, such as a diet high in fiber and plant-based foods, can promote the growth of beneficial gut bacteria and reduce inflammation in the gut. Other approaches, such as probiotics and prebiotics, have beneficial effects on the gut microbiota [20].

*Probiotics*: Probiotics are live microorganisms that confer a health benefit to the host if consumed in an adequate quantity. Probiotics can be found in fermented foods such as yogurt, kefir, kimchi, sauerkraut, and tempeh. The majority of probiotics transiently colonize the gut and are absent in faeces when the intake of probiotics is stopped. For achieving a positive impact on the immune system, these probiotic strains need to colonize the colon for long-term benefits [21].

Prebiotics are non-digestible plant-based dietary fibers that promote the growth and activity of beneficial bacteria in the gut. Prebiotic-rich foods include asparagus, artichokes, bananas, garlic, onions, and whole grains [22].

Other dietary compounds, such as polyphenols can also modulate the gut microbiota. Polyphenol-rich foods include berries, tea, cocoa, and red wine [23]. Additionally, omega-3 fatty acids have anti-inflammatory properties and can promote the growth of beneficial gut bacteria. Omega-3-rich foods include fatty fish such as salmon, mackerel, and sardines [24]. In addition, resistant starch is a type of dietary fiber that is resistant to digestion in the small intestine and reaches the colon intact, where it can be fermented by gut bacteria. Resistant starch-rich foods include cooked and cooled potatoes, green bananas, and legumes [25]. It is important to note that the effectiveness of these natural and dietary sources may vary depending on the individual and the specific cause of gut dysbiosis. In addition to diet, lifestyle factors such as stress management and physical activity can also modulate the gut microbiota. For example, chronic stress has been shown to negatively impact the gut microbiota, while regular exercise has been associated with increased microbial diversity [26].

## 4. Overview of Microbiota

### 4.1. Host Interactions

Microbiota is essential for maintaining the homeostasis of cellular activities by fulfilling metabolic requirements, inhibiting pathogen colonization, and stimulating immune functioning [8]. In the host, there is symbiotic interaction between the microbiota and the immune system.

### 4.2. Intestinal Microbiota

The intestinal microbiota comprises bacteria (*Lactobacillaceae*, *Lactococcus* sp., *Enterococcaceae*, *Bacteroidesthetaiotaomicron*, *Bacteroidesfragilis*, and *Bacteroidesovatus*), viruses, protozoa, and fungi (*Candida*, *Saccharomyces*, *Malassezia*, and *Cladosporium*) [27]. Patients with inborn errors of immunity have abnormal interactions between their microbiome and immune system [28]. These primarily monogenic abnormalities lead to an aberrant inflammatory response, gastrointestinal tract damage, and a higher risk of autoimmune and inflammatory disease development [29].

### 4.3. Lung Microbiome

Diverse microbiotas exist in the respiratory tract, the gut, and the oral cavity, despite being previously thought to be sterile. The oral microbiome and the microbiota of the lungs are comparable. The most frequent phyla detected in healthy lung microbiomes are Firmicutes and Bacteroidetes [30]. Other significant genera include *Prevotella*, *Veillonella*, and *Streptococcus* [30]. The maintenance of immunological tolerance to these commensal bacteria depends on interactions between local microbiota and lung immune cells [31]. During lung inflammation, catecholamines and inflammatory mediators are produced, resulting in the overgrowth of several bacterial species, including *Pseudomonas aeruginosa*, *Streptococcus pneumoniae*, *Staphylococcus aureus*, and *Burkholderia cepacia* complex [32]. Numerous IEIs, including HIES, CVID, IPEX, sIgAD, and DiGeorge syndrome, are associated with respiratory manifestations and pose an increased risk of respiratory tract infections along with the development of asthma [32].

Several factors can contribute to the loss and disappearance of the rich microbiota of the body. The hygiene hypothesis, which contends that people are not being exposed to enough microbial stimulation in utero, is one common explanation. Some of the common causes include the following:

Aging: As people age, the composition and diversity of the microbiome can change, and the abundance of certain beneficial microbes can decrease [33]. This can lead to dysbiosis and an increased risk of age-related health problems.

Antibiotics: Antibiotics can kill both harmful and beneficial bacteria in the body, disrupting the balance of the microbiome and leading to dysbiosis. Repeated or prolonged use of antibiotics can have a particularly significant impact on the microbiome [34].

Diet: A diet high in processed foods, sugar, and saturated fats can promote the growth of harmful bacteria and reduce the abundance of beneficial bacteria in the gut.

Stress: Chronic stress can disrupt the balance of the microbiome and lead to dysbiosis, as well as impair immune function and promote inflammation [35].

Environmental factors: Exposure to pollutants, toxins, and other environmental factors can have a negative impact on the microbiome and contribute to dysbiosis. 

Medical conditions: Certain medical conditions, such as inflammatory bowel disease (IBD), can disrupt the balance of the microbiome and lead to dysbiosis. IEI arise due to genetic anomalies that alter the usual host immunological response or immune control. For patients with impaired immune systems in particular, the microbiome appears to be crucial for maintaining host immunity.

Lifestyle factors: Smoking, excessive alcohol consumption, and lack of exercise can all have a negative impact on the microbiome and contribute to dysbiosis [36].

## 5. Microbiota and Host Interaction in IEI

The major dysbioses associated with the known inborn errors of immunity have been enlisted in Table 1.

Common variable immunodeficiency (CVID): CVID is the most common immune deficiency amongst IEIs in adults with 1 per 25,000 people affected [37]. CVID patients present with recurrent gastrointestinal and pulmonary infections and autoimmunity [38,39,40]. Characteristic immunological alterations include reduced immunoglobulin levels, poor antibody response to vaccines, and reduced switched memory B cells. Recent studies have suggested that complications in CVID are observed due to immune dysregulation attributable to disturbed microbiota [41]. Microbial translocation and dysbiosis may contribute to CVID pathology via epigenetic mechanisms [42]. Phosphatidylinositol-4,5-bisphosphate 3-kinase catalytic subunit delta (*PIK3CD)*, a gene involved in the B cell receptor (BCR) signalling pathway, and other essential genes for B cell function were found to have increased DNA methylation inCVID patients; this hypermethylation hampers naïve to memory B-cell switching [43]. Berbers et al. also suggested the crucial role of epigenetic mechanisms in inflammatory conditions, immune dysregulation, and microbial dysbiosis in CVID [44].

One of the pioneering studies on the microbiome in CVID patients was conducted by Jorgensen et al. and they reported low alpha diversity and reduced genus *Bifidobacterium* levels [45]. Along with this, elevated levels of Clostridia, Bacilli, and *Gammaproteobacteria* species were observed in those patients [46]. Moreover, lipopolysaccharide (LPS) in the systemic circulation of the patients also indicates an enhanced translocation of gram-negative bacteria [47]. Elevated serum endotoxin levels were reported in CVID patients with low serum IgG (<4.9 mg/mL). However, IgA levels were not correlated with endotoxemia. The study indicates that bacterial translocation significantly influences manifestations in CVID [48]. Moreover, the levels of endotoxins could be restored to normal after treatment with IVIg [31].

Shulzhenko et al. reported reduced mucosal IgA in the CVID patients with enteropathy [49]. Three different bacterial taxa, *Acinetobacter baumannii*, *Geobacillus*, and the *otu 15570* bacterium, have been identified as potential contributors to CVID enteropathy. Three distinct bacterial taxa, *Acinetobacter baumannii*, *Geobacillus*, and *otu_15570* bacterium, may contribute to enteropathy in CVID patients. The most common cause of enteropathy in CVID was found to be *Acinetobacter baumannii* [38]. *Geobacillus*, which shares genetic characteristics with segmented filamentous bacteria, induces pro-inflammatory cytokines in a murine model of CVID.

Dietary metabolites may also play a role by altering gut microbiota and inducing systemic inflammation in CVID [50]. Macpherson et al. reported an abundance of Gammaproteobacteria with elevated trimethylamine N-oxide (TMAO) in faeces of CVID patients [50]. TMAO is linked to systemic inflammation (elevated TNF-(α) and IL-12) and may be the consequence of dysbiosis in CVID patients [42]. This supports the notion that treating dysbiosis may reduce the generation of TMAO in patients and possibly decrease systemic inflammation in CVID [50].

Another study discussed the role of fatty acid (FA) imbalance in disrupted gut microbiome in patients with immune diseases [51]. A recent analysis of the plasma fatty acid composition of CVID patients revealed lower concentrations of eicosapentaenoic and docosahexaenoic fatty acids correlated with a lowered anti-inflammatory index [51]. Additionally, it was reported that reduced IgG levels have been associated with a potentially unfavourable FA profile in CVID. Furthermore, enhanced gut microbial diversity has also been associated with high plasma *n*-6 PUFAs, which are altered by rifaximin therapy [52].

Common variable immunodeficiency is the commonest symptomatic inborn error of immunity in adults, and data on the microbiome and its alteration are available in patients with CVID from previously published studies. However, comparing data between previous studies is not straightforward and fallible. This is due to the fact that CVID itself is a heterogenous disease and not a single entity, and criteria used for the diagnosis of CVID may not be uniform in different studies. Therefore, it would be prudent to exercise caution when comparing the microbiome and its diversity across various studies in different cohorts of CVID.

Wiskott-Aldrich syndrome (WAS): WAS is an X-linked disease due to a defect in the *WAS* gene, which manifests with thrombocytopenia, frequent infections, eczema, and increased incidence of malignancy and autoimmunity [53]. Approximately 10% of WAS patients experience microbial dysbiosis, which results in severe, sporadic, and recurring gastrointestinaltract inflammation [54]. Zhang et al. reported a reduced Bacteroidetes and Verrucomicrobia and higher Proteobacteria in WAS patients [55]. WAS patients tend to have an acute and early form of inflammatory bowel disease (IBD) which is phenotypically similar to polygenetic IBD due to microbial dysbiosis [56]. Gingivitis and periodontitis are other common manifestations caused by *Fusobacterium nucleatum*, *Porphyromonas gingivalis*, and *Tannerella forsythia* related to early-onset periodontitis [57]. Although these organisms are a typical component of the microbiota, in a compromised host, these bacteria can cause gingivitis and periodontitis which may lead to premature teeth loss [58].

Severe combined immunodeficiency (SCID): SCID is characterized by a failure in T cell production and function. During the first few months of life, children with SCID frequently get opportunistic infections, fungal, bacterial, or viral, that can be fatal. SCID is regarded as a medical emergency, if diagnosis and effective treatment are delayed afflicted children frequently pass away from severe illnesses.

SCID is a monogenic disorder due to a defect in T-, B-, and NK-cells. T-SCID can be categorized as T-B-NK+, T-B-NK-, T-B+NK-, and T-B+NK+, depending on whether B lymphocytes and natural killer cells are present or absent. Nineteen genes have been implicated in causing the SCID phenotype to date. These include genes *RAG1*, *RAG2*, *IL2RG*, *IL7R*, *JAK3*, *LIG4*, *AK2*, *ADA*, *DCLRE1C*, *FOXN1*, *PRKDC*, *CORO1A*, *CD3D*, *LAT*, *CD3E*, *PTPRC*, and *CD3Z* [59,60]. Patients with SCID have severe life-threatening opportunistic infections early in life with bacteria, fungi, viruses, and protozoa. Depending on the kind of SCID, treatment modalities include hematopoietic stem cell transplantation (HSCT), enzyme replacement therapy, gene therapy, and gene editing [61,62]. These patients do not survive without early management and may present with graft-versus-host-disease (GvHD) following HSCT [63]. GvHD is influenced by multiple factors, including gut microbiota [64]. It was demonstrated that low microbial diversity post-HSCT leads to the emergence of *Escherichia* sp., *Staphylococcus* sp., and *Enterococcus* sp. [6,27,65]. Lane et al. reported changes in gut microbiome composition after HSCT in cases with X-linked SCID and RAG deficiency [66,67]. Another study on patients receiving gene therapy for X-linked SCID observed a reconstitution of the gut microbiota after a normal immune system was established. The effect of the gut microbiome on HSCT and the outcome is also a field of active research [68]. Along with the reconstitution of the immune system, the T-cell receptor repertoire was also normalized [69]. These studies suggest that faecal microbiota transplant (FMT) can be a potential therapeutic option for SCID patients with inflammatory intestinal disease [70].

Chronic granulomatous disease (CGD): Phagocytes are a crucial component of innate immunity; they internalize and kill pathogens and initiate adaptive immune responses [71]. CGD is an oxidative defect disorder in the nicotinamide adenine dinucleotide phosphate oxidase 2 (NOX2) complex, which results in infections (bacterial and fungal) and granulomas [72,73]. These granulomas can interrupt the functions of multiple organs, including the gastrointestinal and urogenital tract [74]. Around 17% of CGD patients have gastrointestinal tract involvement, especially patients with p40phox deficiency who present with intestinal inflammation [75,76]. A study on faecal microbiota composition in patients with CGD showed an abundance of Bacteroidetes and *Clostridiaceae* bacterial families [77]. These microbes are part of the oral microbiome of CGD patients, and their presence in faecal matter indicates their role in gut inflammation [77]. Another study on IBD observed that oral microbiota could colonize the gut and drive Th1-dependent inflammation, thus negatively impacting gut health [78].

IgA Deficiency: The IgA level plays a vital role in mucosal immunity and intestinal diversity and is crucial for balancing the microbiome [79]. IgA provides tolerance to the commensal microbes, expels pathogens, and removes toxins from gut epithelium, promoting gut barrier integrity [80]. Dendritic cells (DCs) can present antigens for neutralization by secretory IgA, which is essential for preserving the mucosal surface’s homeostasis. M-cells in Peyer’s patches of the intestine are critical to performing this function [81]. DCs, epithelial cells, and innate lymphoid cells work predominantly in conjunction with T-cell-dependent and independent pathways to facilitate gut plasma cells for synthesizing IgA [82].

Hyper-IgE syndrome (HIES):HIES is an immunodeficiency disorder with elevated serum IgE levels. Most cases are due to an autosomal dominant (AD) STAT3 deficiency [60]. The patients present with rash, recurrent staphylococcal skin abscesses, pneumonia, chronic mucocutaneous candidiasis (CMC), connective tissue abnormalities, and skeletal anomalies. *STAT3* gene is involved in signal transduction processes for the differentiation of naïve CD4^+^ cells into Th17 cells and proliferation of neutrophils [83]. Defects in STAT3 hamper Th17 differentiation, which enhances the predisposition to fungal (*Candida*) infections. STAT3 triggers the activation of several cytokines, including IL-17 and IL-22. These cytokines produce antimicrobial peptides, which confer protection against staphylococcal atopic dermatitis and CMC [84]. Severe fungal dysbiosis was reported in patients with AD-HIES syndrome compared to healthy individuals [85]. *Candida albicans* predominates over other fungi, such as *C. parapsilosis*, *Boletus*, and *Penicillium*, in patients with AD-HIES [86]. Therefore, defective STAT3 cannot support *Candida albicans* as a commensal organism. *C. albicans* forms a symbiosis in these patients with oral *Streptococcus mutans* and *Streptococcus oralis* [87]. Thus, an impaired STAT3/Th17 axis leads to oral dysbiosis and may allow the *C*. *albicans* to switch from commensal to pathogenic in these patients.

X-linked lymphoproliferative disease type-2 (XLP2): The *XIAP* gene regulates cell survival and inflammatory response, and pathogenic variants in this gene cause X-linked inhibitor of apoptosis protein (XIAP) deficiency, XLP2 [88]. Patients with XIAP deficiency typically show inflammatory symptoms, including Crohn’s-like colitis, and are highly susceptible to hemophagocytic lymphohistiocytosis [40,89]. Severe infectious mononucleosis, splenomegaly, fistulating cutaneous abscesses, and antibody deficiency, which results in recurrent infections, are characteristics of clinical manifestations [90]. Proteobacteria, Firmicutes, Actinobacteria, and Fusobacteria phyla were abundant in patients with XIAP deficiency. Four of these taxa—Fusobacterium, Scardovia, Veillonella, and Rothiadentocariosa—are recognized oral microbiota members. These microbes are typically absent in the gut and have been linked to IBD, colorectal cancer, and liver disorders if present in the gut microbiota [91,92].

Immune dysregulation poly-endocrinopathy enteropathy X-linked (IPEX) syndrome: *FOXP3* gene mutations cause the X-linked IPEX syndrome. The gene produces a crucial transcriptional regulator essential for the maturation and operation of regulatory T-cells (Tregs) [93]. Tregs limit immune system activation and are necessary to avoid systemic autoimmune disease. Tregs in the gastrointestinal tract are generated from conventional FOXP3 neg CD4+ T-cells and induce tolerance to microbiota and dietary antigens [94]. IPEX syndrome should be investigated in male infants who present with eczema, endocrinopathy, dermatitis, and early-onset IBD. Type-1 diabetes mellitus and autoimmune thyroid defect are the commonest endocrinopathies encountered in IPEX syndrome [95]. Studies on the microbiome of Scurfy (SF) mice, which contain a *Foxp3* gene mutation and a clinical presentation resembling IPEX syndrome, revealed a reduced gut microbial diversity [96]. The SF mice had predominantly Bacteroidetes with lower levels of *Lactobacillus* [97]. In contrast, according to Wu et al., patients with severe diarrhea exhibited lower levels of Bacteroidetes and a larger abundance of Firmicutes. Additionally, the adoptive transfer of Foxp3^+^Treg cells to SF mice can restore bacterial diversity [98].

IL10RA deficiency-Patients with IL-10 receptor or IL-10 deficiency typically present with severe IBD in early infancy, frequently accompanied by perianal disease, enterocutaneous lesions, and recto-vaginal fistulas [99]. Patients with Crohn’s disease and IL-10RA deficiency demonstrated reduced gut microbiota diversity [100]. In the IL-10RA group, Firmicutes were found in significantly higher abundance. Intriguingly, the severity of the sickness was directly correlated with the degree of gut dysbiosis in patients with IL-10RA defect. Gut dysbiosis was determined based on the skewed abundance of Lactobacillales, Micrococcales, Veillonellaceae, Clostridiales, and Selenomonadales [101].

Xue et al. observed a reduced variety of gut microorganisms and the preponderance of Firmicutes, Proteobacteria, Actinobacteria, and Bacteroidetes by examining the faecal microbiome composition in the patients with loss-of-function *IL10RA* gene variants. Furthermore, a modest correlation between the severity of the illness and intestinal dysbiosis was found [100,102].
pathogens-12-00518-t001_Table 1Table 1Microbial dysbiosis in various inborn errors of immunity.Inborn Error of ImmunityMicrobiome DysbiosisClinical ImplicationReferencesChronic Granulomatous Disease(CGD)Reduced diversity and abundance of commensal bacteria, overgrowth of opportunistic pathogens *Staphylococcus aureus*Significant abundance of *Proteobacteria* of the *Enterobacteriaceae* family, Bacteroidetes phylum and the *Clostridiaceae* familyNegative health outcomes, including IBD[77,103]Wiskott–Aldrich Syndrome(WAS)Increased abundance of potentially pathogenic Proteobacteria and Roteobacteria, reduced levels of protective commensals such as *Faecalibacterium prausnitzii* and Bacteroidetes and VerrucomicrobiaMay lead to periodontal lesions[55]Severe Combined Immunodeficiency(SCID)Disruption of gut microbiome development and reduced microbial diversity wherein there is increased abundances of *Escherichia*, *Staphylococcus*, and *Enterococcus* as well as *Veillonella*, *Enterobacteriaceae*, Adenovirus and BocavirusIncreased disease severity[6,27,65,66,104]Autoimmune Polyendocrinopathy Syndrome Type 1(APECED)Increased abundance of bacteria associated with autoimmune disorders, such as *Bacteroides fragilis* and *Proteus mirabilis* and reduction in gram-positive FirmicutesSevere gastric symptoms[105]Common Variable Immunodeficiency(CVID)Altered gut microbiota composition, Decreased abundance of beneficial bacteria such as Bifidobacterium and Lactobacillus, *Bacteroides* and Firmicutes and increased abundance of Clostridia, Bacilli, *Prevotella*, and GammaproteobacteriaSystemic inflammation[45,46,49,50]Selective IgA deficiencyIncreased abundance of Firmicutes Bacteroidetes, Gammaproteobacteria and *Prevotella*Systemic inflammation [106,107,108]Hyper-IgE syndrome(HIES)Predominance of *Candida albicans*Decreased abundance of C. *parapsilosis*, *Boletus*, and *Penicillium*STAT3/Th17 axis play an important role in maintaining C. albicans as a commensal organism[87]IL-10 receptor deficiency(IL10R)Increased abundance of Firmicutes, Proteobacteria, Actinobacteria, and BacteroidetesIncreased disease severity[100,109]Immunodysregulation polyendocrinopathy enteropathy X-linked syndrome(IPEX)Significant increase in Bacteroidetes and Firmicutes and a low abundance of *Lactobacillus*FMT represents a promising alternative therapy for severe diarrhea unresponsive to routine therapy.[98]X-linked lymphoproliferative disease type-2(XLP2)Abundance of Proteobacteria, Firmicutes, Actinobacteria, and Fusobacteria (Oral microbiome)Systemic inflammation[92,93]

## 6. Microbiota Modification as Therapeutic Strategy

As the gut microbiota, immune system, and infectious diseases axis is better understood, microbiota modification is increasingly being used as a treatment modality [110]. The gut microbiota is tightly linked with immune response and is essential for maintaining the delicate balance between health and disease. To restore the balance of the gut microbiota and lessen pathogenic activity in various IEIs, prebiotics, probiotics, postbiotics, and fecal microbiota transplantation (FMT) methods are being developed. FMT involves inserting healthy donor faeces into the recipient patient in order to repair the damaged gut flora and deliver therapeutic benefits [85]. Recent studies have reported the use of FMT in immunocompromised patients with recurring and/or resistant *C. difficile* infections (CDI). FMT can potentially be used as a therapeutic choice for mild to moderate ulcerative colitis and is reported to induce remission, treat the dysbiosis linked to chronic liver disease, and eliminate gastrointestinal carriage of antibiotic-resistant microorganisms [111,112]. A recent study observed gut dysbiosis in germ-free mice transplanted with faeces from CVID patients with non-infectious complications but no dysbiosis with FMT from CVID patients with infection-only phenotype or their household contacts. The gut dysbiosis mimicked the derangement seen in non-infectious CVID patients, with an increased abundance of *Dysgonomonas mossii* and *Bacillus massiliensis* [113].

Dysbiosis can be cured by orally taking live, naturally occurring probiotic bacteria [87]. Probiotics promote connections between intestinal epithelial cells, mucosal immune cells, and the gut microbiota, which result in increasing the production of bioactive peptides and safeguarding the intestinal epithelial barriers. There are various studies on the ability of prebiotics to control the gut microbiome and the impact of these agents on human health and well-being. Several preclinical and clinical investigations have demonstrated the immunomodulatory effect of prebiotics. Prebiotic fibers are known to act as a substrate for probiotic commensal bacteria, which release short-chain fatty acids and several other metabolites in the digestive tract. These metabolites utilize G-protein-coupled receptor-mediated pathways to interact with gut-associated epithelial cells as well as local and systemic immune cells. The prebiotics can directly interact with the innate immune cells and gut-associated epithelial cells via Toll-like receptors which produce a microbiota-independent protective effect. This results in a cumulative effect, which helps in maintaining the integrity of the epithelial barrier. In addition, innate immunity is modulated through the release of pro- and anti-inflammatory cytokines, changes in macrophage polarisation and function, neutrophil recruitment and migration, dendritic cell and regulatory T-cell differentiation, and other processes [114]. Prebiotics’ ability to modulate immunity has been used to create potential implications in health as well as supplemental immunomodulatory treatments for a number of IEI [28], using a number of ingredients such as minerals, herbs, other botanicals, amino acids, and enzymes. Common supplements include glucosamine, probiotics, fish oils, minerals like calcium and iron, vitamins D and B12, echinacea, and garlic [114].

He et al. suggested the characterization of microbial flora in IPEX syndrome for developing enhanced therapeutic targets [96]. In the mouse model of IPEX syndrome, the human-derived probiotic *Limosilactobacillus reuteri* DSM 17938 was able to restore the gut microbial equilibrium [115]. This probiotic is used to treat colic episodes and viral diarrhea in infants [116].

High IgG serum levels toward the gut flora have been observed in animal models of innate immune deficiencies. In addition, IgG titers against *E. coli* were documented in IBD patients and secretory IgA-deficient mice [117,118]. Recent mouse studies have revealed that healthy mice manufacture IgG directed towards commensal bacteria under homeostatic settings [119,120]. These systemic IgGs are directed towards conserved antigenic motifs and offer systemic defense against commensal flora. Secretory IgA and systemic IgG confer its effect synergistically to target the gut microbiota. Fadlallah et al. reported enhanced serum anti-microbiota IgG in the patients with selective IgA deficiency (SIgAd) in comparison to healthy controls.It has been speculated that IVIG preparations supplemented with a pool of IgG from patients of SIgAd may improve protection against microbial translocations in CVID patients [108]. Using a B-cell deficient murine model, it was seen that a gluten-free diet (GFD) inhibited the proliferation of bacteria in the small intestine while promoting colonization of a specific pathogen [120]. Falcone et al. demonstrated that ten weeks of exclusive enteral nutrition in a child with XL-CGD disease alleviated IBD and generated long-lasting changes in the microbiome [121,122]. *Faecalibacterium prausnitzii*, which is a major component of healthy gut microbiota and has known anti-inflammatory properties, was also highly enriched in the patient’s gut microbiome [123].

## 7. Conclusions

Host immunity and microbiota interact reciprocally and dynamically. Immune defects affect the microbiota constitution, which may further aggravate the clinical manifestations. IEIs create a unique environment by altering the homeostasis of the protective microbiome, expanding pathogenic microorganisms, or resulting in an ectopic placement of the commensals. There is a need for larger and more focused studies on all aspects of IEIs to understand the microbiome interaction with these immune disorders. Further research would drive advancements in the arena of novel therapeutic options for patients with IEIs.

## Figures and Tables

**Figure 1 pathogens-12-00518-f001:**
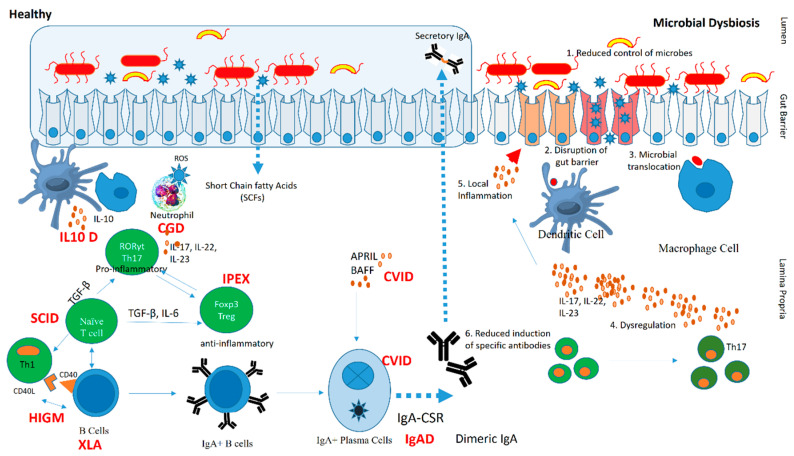
Pathogenic mechanism in microbial dysbiosis due to an underlying inborn error of immunity. The left panel depicts molecular mechanisms in a healthy human host and their alteration in specific IEI (highlighted in red). The right panel shows a mechanism of microbial dysbiosis due to reduced controls of microbes, disruption of the gut barrier, and immune dysregulation. Abbreviations: IL10 Deficiency (IL10D), chronic granulomatous disease (CGD), immune dysregulation, polyendocrinopathy, enteropathy, X-linked syndrome (IPEX), common variable immunedeficiency (CVID), IgA deficiency (IgAD), severe combined immune deficiency (SCID), hyper IgM syndrome (HIGM), and X-linked agammaglobulinemia (XLA).

## Data Availability

Not applicable.

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
