# Peer review of "Microbiome and Its Dysbiosis in Inborn Errors of Immunity"

_pathogens, 2023, doi:10.3390/pathogens12040518_

Round 1

Reviewer 1 Report

This review is quite relevant and interesting. It is very well designed and written as well. It is also written with relevant contents. But, here, it needs to add few more well justified and required contents, which lies in the scope of this study title. It will make the review more conclusive and enhance the scope as well. Here, I have suggested few comments that may be addressed and added in this current review.

1.     The section “modification as therapeutic strategy” may be enlarged with quoting and mentioning more option available as natural sources or other dietary sources. It will further strength and enhanced the scope of this review.

2.     Microbiota ensure the healthiness of the body through its involvement in different metabolic pathway. Moreover, a disruption and disappearing in the body's microbiota is well reported in various study.  Hence, it is very essential to mentioned the common cause of the loss and disappearance of the rich microbiota of the body.

3.     How the diet and other natural ways may modulate the rich Microbiota within the body, may be added as a new section in this review.

4.     Aging is the common cause of loss of microbiota.  How this could be checked and regulated? A section dealing “Aging and Microbiota” may be added, as well.

Author Response

S No

Reviewer’s Comments

Response to reviewer’'s’ comments

Reviewer 1

1.

The section “modification as a therapeutic strategy” may be enlarged with quoting and mentioning more option available as natural sources or other dietary sources. It will further strength and enhanced the scope of this review.

We are thankful to the reviewer for their valuable suggestion.

As per the suggestion, we have incorporated the changes in the revised manuscript on page 7 lines 289-324.

2.

Microbiota ensures the healthiness of the body through its involvement in different metabolic pathway. Moreover, a disruption and disappearing in the body's microbiota is well reported in various study.  Hence, it is very essential to mentioned the common cause of the loss and disappearance of the rich microbiota of the body.

We are thankful to the reviewer for their valuable inputs and comments.

We have incorporated the changes in the revised manuscript on pages 3-4 lines 121-143

3.

How the diet and other natural ways may modulate the rich Microbiota within the body, may be added as a new section in this review.

Necessary changes have been incorporated in the revised manuscript for your reference.

4.

Aging is the common cause of loss of microbiota.  How this could be checked and regulated? A section dealing “Aging and Microbiota” may be added, as well.

We thank the reviewer for the observation.

We would like to emphasise that the manuscript already have a subsection with the heading “Normal microbiome at different ages, from newborn to adult” and as per your suggestion we have changed the heading of the same to Aging and Microbiota in the revised manuscript and expanded the section as advised.

Reviewer 2 Report

The topic is of great interest. However, this article does not add relevant details to the available literature.

Please see the following paper: Gut Microbiota-Host Interactions in Inborn Errors of Immunity. Int J Mol Sci. 2021 Jan 31;22(3):1416. doi: 10.3390/ijms22031416.

I find an important overlap here. It would be interesting if the Authors could modify the paper, adding the most recent data published after the above-cited article.

If the Authors include only novel studies (specifying it is an addition to the above-cited paper), then the paper can really add to the literature. 

Author Response

Reviewer 2

1.

The topic is of great interest. However, this article does not add relevant details to the available literature.

Please see the following paper: Gut Microbiota-Host Interactions in Inborn Errors of Immunity. Int J Mol Sci. 2021 Jan 31;22(3):1416. doi: 10.3390/ijms22031416.

I find an important overlap here. It would be interesting if the Authors could modify the paper, adding the most recent data published after the above-cited article.

If the Authors include only novel studies (specifying it is an addition to the above-cited paper), then the paper can really add to the literature.

We would like to thank the reviewer for their valuable suggestion.

We have done an extensive literature search post this review article and to the best of our knowledge we have incorporated all the studies available till date in the revised manuscript cited at (113, 124)

Reviewer 3 Report

The authors have prepared a very nice review on a topic of considerable interest, i.e. the interplay between immune deficiency and the microbiota.  In general, the paper is well written and easy to digest.  I have three major comments.

First, it would be helpful to prepare a table that lists the IEI and the most notable known changes in the microbiota, along with potential clinical implications.  This would be very helpful for both investigators and practicing clinicians as it would provide an outline for approaching these patients.

 Second, as noted by the authors, the microbiota reflects the immune status of the individual and the local biodiversity of the microbiota.  Although this is mentioned at the beginning of the review, there are no comments regarding the effects of the local environment in the context of immune deficiency and the microbiome.  This information may not be available, but if so this should be mentioned.

Third, although most of the immune deficiencies discussed are the product of single gene mutations, CVID is a minefield. 

First we begin with the diagnosis of CVID.  What you have to take into account is the difference between a purist academic view of the diagnosis of common variable immune deficiency, which is helpful in writing papers and comparing patients across facilities, and a real world experience with the spectrum of disease that we lump into adult primary antibody deficiency.  To meet the purist’s standard, CVID requires an IgG of less than 500 mg/dL with either an IgM or IgA level that is two standard deviations below normal, and some evidence of difficulties responding to bacterial polysaccharides (e.g., response to Pneumovax) in the presence of normal numbers of T cells and B cells. 

In the field, however, there is clearly a spectrum of disease for which patients meeting the diagnosis criteria for CVID are a minority.   The largest categories are

·       CVID, meeting the criteria above;

·       hypogammaglobulinemia (IgG deficiency, almost always with a low IgG1 with or without a low IgG2) with an IgG of less than 500 mg/dL and normal IgM and IgA with moderately or severely impaired responses to Pneumovax;

·       IgG2 subclass deficiency with or without IgA or IgG4 subclass deficiency with moderately or severely impaired responses to Pneumovax;

·       an intermediate category (ICR for intermediate CVID/recurrent sinopulmonary infections) where the IgG is between 500 and 600 mg/dL with a low IgA or IgM and moderately or severely impaired responses to Pneumovax; and

·       specific antibody deficiency (SAD) where the responses to Pneumovax are moderately or severely impaired but serum Ig levels are normal. 

In all of these cases, T cell numbers are often low but sometimes high; and B cell numbers are typically on the low end of normal or below; but present. 

Patients with CVID are a minority of this spectrum, but as noted above, include both patients with IgA as long as they lack IgM.  Also, complete absence of IgA is rare.  Most patients have detectable IgA levels even when they are considered IgA deficient. 

Commonly, most CVID patients, as well as patients with hypogammaglobulinemia (IgG deficiency) alone complain of GERD and low level dyspepsia, whether they are on antibiotics or not, or on immunoglobulin replacement therapy or not.

Less than ¼ of patients with CVID had a recognizable single gene variant that can be associated with their disease, and penetrance and expressivity varies.  The average age of onset is in the 30’s to 40’s. 

These facets make create issues in discussions of the range of deficits seen in CVID, especially when the criteria used to identify patients as having CVID can vary.  Adding further difficulty is that authors of manuscripts describing clinical differences in patients with CVID may not provide the criteria they used to define their patient populations. 

This is perhaps a more complex discussion than would be warranted for a review of this type.  Still, these issues need to be kept in mind when writing the CVID section.  As a limited example, lined 140 and 141 need to be clarified.  “Elevated endotoxin levels in serum, and low IgA levels have also been reported in CVID, however, no association was found between IgG and IgM levels [31].”  Does this mean that elevated endotoxin levels are found only in CVID patients that lack IgA, or not?

Author Response

Reviewer 3

1.

First, it would be helpful to prepare a table that lists the IEI and the most notable known changes in the microbiota, along with potential clinical implications.  This would be very helpful for both investigators and practicing clinicians as it would provide an outline for approaching these patients.

As per the suggestion we have incorporated a table describing the various IEIs and the dysbiosis observed in a tabular form (Table:1).

2.

Second, as noted by the authors, the microbiota reflects the immune status of the individual and the local biodiversity of the microbiota.  Although this is mentioned at the beginning of the review, there are no comments regarding the effects of the local environment in the context of immune deficiency and the microbiome.  This information may not be available, but if so this should be mentioned.

We thank the reviewer.

We have done literature search and the same has been added in the revised manuscript

3.

Third, although most of the immune deficiencies discussed are the product of single gene mutations, CVID is a minefield. First, we begin with the diagnosis of CVID.  What you have to take into account is the difference between a purist academic view of the diagnosis of common variable immune deficiency, which is helpful in writing papers and comparing patients across facilities, and a real world experience with the spectrum of disease that we lump into adult primary antibody deficiency. Adding further difficulty is that authors of manuscripts describing clinical differences in patients with CVID may not provide the criteria they used to define their patient populations.  

We have now explicitly mentioned and acknowledged difficulties and challenges in interpreting data pertaining to microbiome and its dysbiosis in common variable immunodeficiency due to profound heterogeneity of the disease itself and the variable criteria which are used to categorize patients under the rubric of “common variable immunodeficiency” in different studies.

Common variable immunodeficiency is the commonest symptomatic inborn error of immunity in adults and data on microbiome and its alteration are available in patients with CVID from previously published studies. However, comparing data between previous studies is not straightforward and fallible. This is due to the fact that CVID itself is heterogenous disease not a single entity and criteria used for diagnosis of CVID may not be uniform in different studies. Therefore, it would be prudent to exercise caution when comparing microbiome, its diversity and differences across different studies in different cohorts of CVID.

4.

As a limited example, lined 140 and 141 need to be clarified.  “Elevated endotoxin levels in serum, and low IgA levels have also been reported in CVID, however, no association was found between IgG and IgM levels [31].”  Does this mean that elevated endotoxin levels are found only in CVID patients that lack IgA, or not?

The paper published by Perreau et al., has reported elevated endotoxin levels in all CVID patients with low IgG. However, IgA levels were not correlated with endotoxin levels. The sentence has now been corrected.

Round 2

Reviewer 1 Report

Dear Author,

The current modified version is looking more inclusive and relevant. The suggested comments were addressed sufficiently. But, it was very tough for  me to find the incorporation/modification made by you, for my comments suggested. As, the line number you quoted in "Response to reviewer’s comments" is not matching with modified version. e.g. Address for my first comments were nowhere on page 7 lines 289-324 in revised manuscript. It was same for others as well.

Thanks

Reviewer 2 Report

The Authors addresses my comments.

Reviewer 3 Report

The authors have responded to my concerns.